# Advances in the Treatment and Prevention of Chemotherapy-Induced Ovarian Toxicity

**DOI:** 10.3390/ijms21207792

**Published:** 2020-10-21

**Authors:** Hyun-Woong Cho, Sanghoon Lee, Kyung-Jin Min, Jin Hwa Hong, Jae Yun Song, Jae Kwan Lee, Nak Woo Lee, Tak Kim

**Affiliations:** Department of Obstetrics and Gynecology, Korea University College of Medicine, Seoul 02841, Korea; limpcho82@korea.ac.kr (H.-W.C.); mikji@naver.com (K.-J.M.); jhhong93@korea.ac.kr (J.H.H.); sjyuni105@gmail.com (J.Y.S.); jklee38@korea.ac.kr (J.K.L.); nwlee@korea.ac.kr (N.W.L.); tkim@kumc.or.kr (T.K.)

**Keywords:** gonadotoxicity, fertility preservation, embryo cryopreservation, oocyte cryopreservation, ovarian tissue cryopreservation, oocyte in vitro maturation, artificial ovaries, stem cell technologies, ovarian suppression, oncofertility

## Abstract

Due to improvements in chemotherapeutic agents, cancer treatment efficacy and cancer patient survival rates have greatly improved, but unfortunately gonadal damage remains a major complication. Gonadotoxic chemotherapy, including alkylating agents during reproductive age, can lead to iatrogenic premature ovarian insufficiency (POI), and loss of fertility. In recent years, the demand for fertility preservation has increased dramatically among female cancer patients. Currently, embryo and oocyte cryopreservation are the only established options for fertility preservation in women. However, there is growing evidence for other experimental techniques including ovarian tissue cryopreservation, oocyte in vitro maturation, artificial ovaries, stem cell technologies, and ovarian suppression. To prevent fertility loss in women with cancer, individualized fertility preservation options including established and experimental techniques that take into consideration the patient’s age, marital status, chemotherapy regimen, and the possibility of treatment delay should be provided. In addition, effective multidisciplinary oncofertility strategies that involve a highly skilled and experienced oncofertility team consisting of medical oncologists, gynecologists, reproductive biologists, surgical oncologists, patient care coordinators, and research scientists are necessary to provide cancer patients with high-quality care.

## 1. Introduction

Cancer incidence is rapidly growing worldwide. In 2018, 8.6 million women were diagnosed with cancer globally [1]. Most women diagnosed with cancer are older, but 10% are <45 years of age [2]. Due to advances in cancer diagnosis and treatment, the survival rate for prepubertal and young women with cancer has significantly improved. In Europe, the five-year-survival rate is 79.1% in children diagnosed with cancer [3]. However, aggressive chemotherapy can cause impairment of reproductive functions and even fertility loss [4,5,6,7]. Although depletion of ovarian function is associated with improved survival outcomes in breast cancer patients of reproductive age, it has several side effects, such as hot flashes, osteoporosis, and sexual dysfunction [8]. Cardiovascular disease is the main cause of shortened life expectancy in women with premature ovarian insufficiency (POI) [9]. Moreover, chemotherapy-related POI and infertility may be associated with increased risk of neuro-degenerating disease and psychosocial distress [9].

In recent years, interest in fertility preservation has increased significantly among female cancer patients [10]. Despite the huge interest cancer patients have with respect to preserving fertility, there is an unmet need in children and young cancer survivors [11]. Oncofertility is a relatively innovative concept that describes a multidisciplinary network of experts focused on developing and providing the option of fertility preservation to young cancer patients. Currently, embryo and oocyte cryopreservation are the only established methods for fertility preservation [12]. However, there is accumulating evidence for other experimental techniques including ovarian tissue cryopreservation, artificial ovaries, and in vitro maturation [13].

This review will focus on current challenges and future directions to treat and prevent chemotherapy-induced infertility in girls and young women with cancer. We also address current knowledge on chemotherapy-induced ovarian toxicity and its mechanisms.

## 2. The Effect of Chemotherapy on Ovarian Function

### 2.1. Risk of Ovarian Toxicity Due to Chemotherapy Agents

Although the survival rate of cancer patients has dramatically improved due to development of chemotherapy, ovarian toxicity induced by chemotherapy is a major complication. Gonadotoxic chemotherapy during reproductive age can lead to iatrogenic primary ovarian insufficiency (POI), and loss of follicular reserve that depends on the type, dose, duration, and combination of chemotherapeutic agents, and disease stage, as well as patient age [14,15]. It has been reported that 53–89% of chemotherapy-induced POI occurs in patients with breast cancer [16]. The combination of abdominal radiation and alkylating agents which are likely to cause gonadotoxicity induces POI in almost 100% of cancer patients [17,18]. In a large study of cancer survivors, the risk of POI was increased 9.2-fold for patients who received chemotherapy including alkylating agents and 27-fold in women who received combination alkylating agent-based chemotherapy and radiotherapy [19].

Figure 1 and Table 1 show the most common cancers and the risk of chemotherapy-induced ovarian toxicity in women according to chemotherapy protocol and age [3,20,21]. Notice that the course of chemotherapy and its related risks of gonadotoxicity can be unpredictable and variable due to treatment response and disease prognosis, i.e., refractory or recurrent cases [13,22]. In order to prevent POI due to chemotherapy and subsequent complications, effective and comprehensive oncofertility strategies should be undertaken to preserve fertility in young reproductive age women before initiation of cancer treatment [17,23,24,25,26].

### 2.2. Mechanisms of Ovarian Toxicity

Gonadotoxic chemotherapy leads to primordial follicle loss, resulting in POI and infertility. Both direct acute and indirect delayed mechanisms have been reported for the effects of anticancer agents that cause a decrease in ovarian reserve. The main mechanism is that anticancer drugs directly induce DNA double-strand breaks (DSBs), which activate apoptosis and/or autophagy-related pathways [36,37,38,39,40,41,42]. The second mechanism is that anticancer drugs can indirectly cause primordial follicle depletion by microvascular and stromal injury through ischemia, necrosis, or inflammation [38,42,43,44,45]. There is third hypothesis called the “burnout” effect. A few studies have shown that anticancer drugs induce activation of the phosphoinositide 3-kinase/protein kinase B/forkhead box protein O3a (PI3K/AKT/FOXO3a) pathway, which leads to follicle reduction by massive activation of primordial follicles in mice and cultured human ovarian tissue [36,46,47,48,49]. However, there is some question of methodology and biological mechanism of follicle loss based on studies supporting “burnout theory”. It has not been proven that primordial follicle growth is the main cause of chemotherapy-induced primordial follicle loss. Thus, the “burnout” theory of chemotherapy-induced follicle depletion is still lacking evidence and is under debate [36,46]. The main cause of the follicle depletion induced by chemotherapy seems to be DNA double-strand breaks and apoptosis.

## 3. Fertility Preservation Options

Fertility preservation options for women should consider patient age, marital status, chemotherapy regimen, economic status of patients, cancer type, staging upon diagnosis, and the possibility of treatment delay. In addition, whether the cancer is hematological or solid should be evaluated as cancer cells may be present in ovarian tissue, affecting fertility preservation plans. Several methods for fertility preservation in females have been introduced, including embryo and oocyte cryopreservation, ovarian tissue cryopreservation, oocyte in vitro maturation, artificial ovaries, stem cell technologies, and ovarian suppression (Table 2).

### 3.1. Embryo Cryopreservation

Embryo cryopreservation is the gold standard method and has been widely used worldwide for decades. Currently, the transfer of frozen–thawed embryos is as effective as fresh embryo transfer in terms of pregnancy rate [50]. In addition, observational studies and systemic reviews have suggested that frozen–thawed embryo transfer is superior to fresh embryo transfer in terms of clinical outcomes [51]. Although there is concern about the effects of storage duration on frozen embryos, several studies have shown that the duration of cryopreserved embryo storage had no negative effects on pregnancy or live birth rate [52,53,54,55,56]. Embryo cryopreservation consists of ovarian stimulation, mature oocyte retrieval, and in vitro fertilization (IVF) with sperm. For embryo freezing, there are two methods: slow freezing and vitrification. Several studies, including a recent meta-analysis, suggested that the embryo vitrification and thawing method is better than slow freezing and thawing in terms of pregnancy and live birth rates [57,58,59]. Because this technique requires ovarian stimulation, it is not suitable for prepuberal girls or women who do not have a partner or do not want sperm donation. In estrogen-dependent cancer, such as breast or endometrial cancer, ovarian stimulation is not suitable because it can increase blood estrogen levels, but alternative ovarian stimulation protocols with aromatase inhibitors can be used [60,61]. In addition, it is difficult to apply this method in aggressive cancer that requires immediate treatment, as chemotherapy may be delayed due to possible adverse effects such as ovarian hyperstimulation syndrome (OHSS). For these patients, random-start ovarian stimulation may be an alternative [62,63,64,65].

It is known that the pregnancy rate per cryopreserved embryo is 30–35%, and the risk of congenital anomaly is not increased [66,67,68,69]. One retrospective study demonstrated that pregnancy rate per transfer for cancer patients was similar to patients who underwent IVF due to tubal factor infertility (37% versus 43%, *p* = 0.49) and live birth rate per transfer was also was not significantly different (30% versus 32%, *p* = 0.85) [70]. In addition, studies that compared IVF and embryo cryopreservation in cancer patients to those without cancer did not show significant differences in the number of harvested oocytes, fertilization rate, and live birth rate, although there were few good quality embryos in patients with cancer [71,72].

### 3.2. Oocyte Cryopreservation

Oocyte cryopreservation, like embryo cryopreservation, is considered a gold standard technique for fertility preservation in cancer patients [15,73]. In 2000, the United Kingdom (UK) Human Fertilization and Embryology Authority (HFEA) allowed the use of frozen oocytes for fertility preservation [74]. Subsequently, in 2013, the American Society for Reproductive Medicine (ASRM) declared that oocyte cryopreservation was no longer an experimental technique based on four clinical trials [75,76,77,78].

Oocyte freezing involves ovarian stimulation and mature oocyte cryopreservation. Therefore, this technique is not feasible for prepubertal girls. In addition, this has similar disadvantages seen in embryo preservation because it requires ovarian stimulation to obtain a mature oocyte. On the other hand, it is suitable for single women who do not want sperm donation or embryo freezing.

Several previous studies have suggested that vitrification is better than slow freezing in oocyte cryopreservation. Data from a meta-analysis suggested that pregnancy rates associated with frozen oocytes could be improved with the use of vitrification [71]. Subsequent studies have reported that vitrified oocytes show better survival, fertilization, and pregnancy rates than slow-frozen oocytes [79,80,81]. Based on increasing evidence, a 2013 update to the National Institute for Health and Care Excellence (NICE) guidelines recommends vitrification instead of controlled-rate freezing for cryopreservation of oocytes and embryos given the availability of necessary equipment and expertise [82].

According to randomized controlled trials, pregnancy rates per frozen–thawed oocyte were not significantly different from IVF using fresh oocytes (from 4.5 to 12%) [75,76,77,78]. Additionally, there was no increase in congenital birth defects or developmental delays in children born from oocyte cryopreservation [83].

Because oocyte freezing involves removal of cumulus cells before cryopreservation, it may cause changes in the zona pellucida that could lower fertilization rates of conventional insemination [84]. Therefore, Practice Committees of the ASRM and the Society for Assisted Reproductive Technology recommend intracytoplasmic sperm injection (ICSI) for frozen oocytes as the preferred method for achieving fertilization, although there are limited data to support this technique [85]. Similar to embryo freezing, this technique involves concerns and uncertainties regarding the efficacy and long-term effects, although one study has shown that long-term cryopreservation of oocytes has no significant negative effects on live birth outcomes [86].

### 3.3. Embryo vs. Oocyte Cryopreservation

Compared with oocyte cryopreservation, embryo freezing has several advantages, including that storing excessive embryos can reduce the risk of multiple pregnancy by reducing the number of embryos transferred and can increase cumulative pregnancy rates. However, embryo cryopreservation requires a male partner or use of donor sperm, which can raise ethical and legal concerns [87]. Oocyte cryopreservation can provide women autonomy regarding reproduction. Just as embryo cryopreservation is not considered an alternative to sperm freezing in male fertility preservation, this option should be approached carefully in terms of women’s rights, as oocyte cryopreservation has been accepted as a standard process for fertility in today’s world of assisted reproduction.

### 3.4. Ovarian Tissue Cryopreservation and Transplantation

Although ovarian tissue cryopreservation is considered experimental, it has several advantages compared to embryo or oocyte cryopreservation. First, it is the only option for fertility preservation in children, adolescents, and young adult cancer patients who need immediate chemotherapy and do not have sufficient time for ovulation induction. Second, the procedure can be performed regardless of menstrual cycle stage. Third, a large number of oocytes including primordial follicles can be preserved. Fourth, the hormonal function of the ovary can be restored, which improves the quality of life for young women. Finally, this technique does not need ovarian stimulation or a sperm donor [15,88,89,90].

This method involves laparoscopic ovary excision and freezing for preservation before initiation of chemotherapy. There are three options for ovarian tissue excision, which are (1) ovarian cortex biopsy, (2) partial oophorectomy, and (3) complete oophorectomy [91]. According to von Wolf’s group, 50% resection of the ovary may be sufficient for cryopreservation [92]. For ovarian tissue freezing, vitrification has been attempted in several recent studies with promising results [93,94,95,96,97]. However, these results are not enough to recommend the vitrification method to patients. Donnez et al. reported that the first 24 live births from human ovarian tissue cryopreservation and transplantation were achieved by slow-freezing methods [98]. Until now, slow freezing has been considered to be a more suitable method for ovarian tissue cryopreservation than vitrification [99].

After cancer remission is achieved, frozen ovarian sections are thawed and implanted on the surface of the remaining ovary or on the peritoneum [100]. In most cases, frozen–thawed ovarian tissue is orthotopically transplanted, but if orthotopic transplantation is not possible, it can be heterotopically transplanted to other areas, such as the subcutaneous space of the abdominal wall or forearm [13].

According to data from published research, since the first pregnancy was reported using this technique in 2004, the number of live births after ovarian tissue slow freezing and orthotopic auto-transplantation has exceeded 120 [13,101,102,103,104,105,106,107,108]. According to data from five major centers, the pregnancy and live birth rates were 29% and 23%, respectively [108]. In a subsequent case series of 74 women, the pregnancy rate was 33% and live birth rate was 25% [107]. According to data from Donnez’s group, pregnancy and live birth rates were 41% and 36% in 22 women who underwent ovarian tissue cryopreservation and auto-transplantation [10]. Patient age at the time of cryopreservation is a major predictive factor affecting improved future pregnancy outcomes [15]. In general, an age of 35 years is regarded as the upper limit for ovarian tissue freezing, because primordial follicle count significantly decreases with age [109].

A review of 60 cases of re-implantation showed ovarian activity was restored in 92.9% of cases after transplantation of cryopreserved ovarian tissue by the slow-freezing method [98]. It has been reported that approximately 3.5 to 6.5 months were required for an increase in estradiol and a decrease in FSH levels. After ovarian tissue transplantation, restoration of ovarian function has been reported consistently, along with an increasing number of successful live births [15].

Although ovarian tissue cryopreservation may be feasible in patients with aggressive cancer requiring immediate chemotherapy, possible contamination of ovarian tissue with malignant cells is a major concern associated with this technique [13,110]. Low levels of malignant cells have been detected in ovarian tissue and can lead to recurrence of disease after transplantation in both mouse and human models with hematologic malignancies [111,112,113]. Therefore, this technique is contraindicated in women with ovarian or hematologic malignancies. In these patients, in vitro maturation of oocytes and artificial ovary technology should be considered to preserve and restore fertility [13]. Some researchers have suggested that ovarian tissue can be cryopreserved after an initial course of chemotherapy to reduce the risk of cancer contamination despite damaging ovarian function [105,114].

### 3.5. In Vitro Maturation (IVM) of Oocytes

Recently, IVM has been widely applied to immature oocytes that have been collected from women with polycystic ovarian syndrome (PCOS). However, there are still several controversies regarding the application of IVM in the oncofertility field [115]. In subfertile women with PCOS who need assisted reproductive techniques including ovarian hyperstimulation and IVF, IVM has been suggested to prevent or overcome complications such as OHSS and retrieval of immature oocytes [116]. In these cases, this method involves the in vitro culture of immature oocytes until the metaphase II stage. When this technique is applied to women with cancer, IVM involves an in vitro culture of fresh or frozen–thawed ovarian tissue, isolation of ovarian follicles, or immature oocytes for maturation into metaphase II oocytes for further IVF [117]. This method does not require an ovarian stimulation process. Therefore, it is feasible for all patients including prepubertal girls and patients who need to receive immediate chemotherapy.

Over 4000 babies have been born by assisted reproductive technology (ART) using IVM, mainly in women with PCOS and no increase in congenital defects or developmental delays due to IVM have been reported [118,119]. There are now a number of case reports of IVM success and live births in women with cancer, using IVM culture following ex vivo collection of immature oocytes after an oophorectomy [120,121].

### 3.6. Artificial Ovary

Artificial ovary is an experimental method to obtain mature oocytes by an ex vivo multistep procedure involving in vitro cultures of ovarian tissue, follicles, and oocytes [122]. According to previous research, ovarian follicles and their enclosed oocytes can be harvested before or after ovarian tissue freezing and thawing [123,124,125,126,127,128,129]. Although this technique is still challenging in humans, it has shown promising results in animal models [130,131]. Further research advances and successes will improve the results of this technique and it may offer another safe option for preserving fertility in women with cancer.

The applications for artificial ovaries include three options. The first option is in vitro fertilization and/or vitrification of in vitro matured oocytes [132,133,134,135,136,137,138,139,140]. A few recent studies show that the live birth rate through this option is comparable with traditional IVF [133,136]. The second is retransplantation of in vitro activated ovarian tissue. This option was successful in women with premature ovarian failure and resulted in heathy live births in some cases [128,141,142,143]. Finally, in vitro-grown ovarian follicles can be retransplanted in a 3D biodegradable microenvironment. Although many studies have demonstrated that this technique can be performed in animal models, there has been no human trial to date [123,124,128,129,141,142,143].

### 3.7. Gonadotropin-Releasing Hormone (GnRH) Analog

It has been suggested that ovarian suppression by administration of a GnRH analog before or during chemotherapy may have protective effects on ovaries by down-regulation of FSH and pituitary LH secretion [144]. There are two hypotheses for the mechanism of GnRH analogs [145,146,147]. One hypothesis is that the primordial follicles entering the growing pool decreases, resulting in decreased sensitivity to gonadotoxicity caused by administration of GnRH analogs. Another hypothesis is that GnRH analogs may have a direct antiapoptotic effect on ovarian germline stem cells. However, it is difficult to explain why GnRH analogs have no protective effect on ovaries after radiotherapy [3,12,21,148,149,150,151,152]. A recent randomized trial demonstrated that a goserelin + chemotherapy group had fewer cases of POI and more successful pregnancies without adverse effects [153]. Although several previous studies that included a randomized trial and meta-analysis demonstrated that temporary ovary suppression by GnRH analogs could reduce chemotherapy-induced gonadotoxicity, the protective effect and mechanisms of GnRH analogs on ovaries are unclear and widely debated [154,155,156,157,158,159,160,161]. According to the 2018 American Society of Clinical Oncology (ASCO) Clinical Practice Guideline Update, there is conflicting evidence on recommending GnRH analogs and other means of ovarian suppression for fertility preservation [73]. When established fertility preservation options are not feasible in young women with cancer, such as hormone receptor-positive breast cancer, a GnRH agonist may be offered to reduce chemotherapy-induced ovarian insufficiency. In addition, GnRH agonists can be used as a combination strategy with proven fertility preservation methods such as oocyte or embryo freezing as a safer option compared to GnRH agonist alone [162]. However, GnRH agonists should not be used as the only option if proven fertility preservation methods are available [73].

### 3.8. Ovarian Stem Cells

Recent studies in stem cell research have investigated the application of ovarian stem cell use in fertility preservation. Tilly et al. reported the successful detection and isolation of ovarian stem cells (OSCs) from animal and human ovaries. In subsequent studies, researchers observed these cells giving rise to young egg cells or oocytes, which may hold the key to better treatments for female infertility [163,164,165,166,167,168,169,170]. OSCs obtained from mice can differentiate into oocytes in vitro and are suitable to be fertilized and implanted in animal models and result in embryo development [169]. This may become an option for prepubertal children with cancer and women with different infertility conditions. However, because there is no evidence from clinical use or trials of OCS application for fertility preservation, it is still challenging to use this technique in routine clinical practice, especially in cancer patients.

## 4. Improving Oncofertility Care

Preserving fertility is important to most young cancer survivors. One study reported that more than half (51.7%) of young women undergoing cancer treatment felt that having children was the “most important” issue in their life [171]. The fear of treatment-related infertility may affect patients’ decision making in choosing cancer treatment among those who want to conceive their own genetic offspring [172,173]. Therefore, according to the ASCO, clinicians should refer cancer patients who are undecided or uncertain about their fertility intentions to a reproductive specialist for a fertility preservation consultation before initiating cancer treatment [61,73]. Established in 2007, the Oncofertility Consortium (OC) is a nationwide network of oncologists, reproductive specialists, and research scientists for fertility preservation in young cancer patients [174]. The National Physicians Cooperative, formed by the OC to share knowledge and resources, comprises 60 centers across the United States that provide oncofertility services to women [148]. In Japan, after establishment of the Japan Society for Fertility Preservation (JSFP) in 2012, there are 46 current medical institutions for preserving fertility. In Europe, the FertiPROTEKT network was founded in May 2006, and has included approximately 100 centers from Germany, Austria, and Switzerland since January 2014 [174].

Despite the increasing interest in and the advance of technologies available in the oncofertility field, accessibility to fertility preservation remains relatively low for young cancer patients, particularly those in low- and middle-income countries [175,176]. In a retrospective cohort study of women aged 18–42 years diagnosed with cancer, 20.6% received fertility preservation care [175]. In another study, only 9% of patients received any information on fertility preservation options [177].

Major barriers are lack of awareness among oncologists, lack of referrals from oncologists, lack of interinstitutional networks, and lack of oncofertility specialists [20]. In addition, many oncologists fail to have fertility discussions with their cancer patients and, thus, fail to make timely referrals due to patients’ lack of awareness of treatment-related infertility, together with time pressures, financial costs, and conflicting priorities of physicians [165,178].

To provide fertility preservation strategies to prepubertal and young women with cancer, each medical institution should be properly equipped, and should have a highly skilled and experienced oncofertility team which consists of medical oncologists, gynecologists, reproductive biologists, oncologic surgeons, patient navigators, and research scientists [13]. When oncofertility care is not available in institutions that treat women with cancer, immediate referral of patients to specialized oncofertility centers is encouraged to assure a high standard of care. In addition, individualized fertility preservation options should be considered based on patient age, marital status, economic status of patients, cancer type, staging upon diagnosis, chemotherapy regimen, and urgency of chemotherapy treatment (Figure 2).

## 5. Other Considerations for Fertility Preservation

### 5.1. Emergency Fertility Preservation

If neoadjuvant chemotherapy is needed or if chemotherapy cannot be delayed due to aggressiveness of the disease, several therapeutic strategies for female oncofertility can be suggested. If it is difficult to delay chemotherapy for about two weeks, such as for leukemia, ovarian tissue cryopreservation can be considered as a fertility preservation option. After completion of cancer treatment, frozen ovaries can be used for ovarian tissue transplantation or in vitro maturation. If chemotherapy can be delayed for about two weeks, random-start ovarian stimulation and subsequent embryo or oocyte cryopreservation may be an alternative. In addition, GnRH agonist administration before or during chemotherapy can be considered for these patients to reduce ovarian toxicity due to chemotherapy.

### 5.2. Timing of Conception after Cancer Treatment

Cancer survivors who want to have children after cancer treatment wonder when they safely can become pregnant. Many physicians and organizations suggest that women postpone pregnancy for 6 to 12 months after finishing chemotherapy to prevent conception with an oocyte that was maturing during chemotherapy [179]. In young women with estrogen receptor-positive (ER+) breast cancer, because adjuvant anti-estrogen therapy is required for 5–10 years, it can lead to a delay of childbearing [180]. However, there is insufficient information about when it is safe to become pregnant after treatment for cancer. An Australian population-based study suggested that waiting at least 2 years after diagnosis to attempt conception is associated with improvement of offspring survival outcomes [181]. Decisions should be made through a multidisciplinary system consisting of the patient, oncology team, and fertility specialist. In particular, in hormone receptor-positive breast cancer patients, adjuvant anti-estrogen therapy can be stopped, and pregnancy may be attempted. Most patients with hormone-positive breast cancer receive anti-estrogen therapy such as tamoxifen or aromatase inhibitors, and follow-up is required every 3–6 months in these patients due to potential risk of endometrial hyperplasia. In addition, the serum anti-Müllerian hormone level should be assessed to evaluate ovarian reserve if the patient wishes to become pregnant.

### 5.3. The Psychosocial Aspect of Fertility Preservation

When considering fertility preservation options, the emotional issues that arise in fertility preservation patients should be evaluated along with the medical safety and efficacy of preserving fertility strategy. Psychosocial factors such as anxiety about recurrence or mortality of disease and uncertainty of fertility-preserving treatment can influence patient decision-making about fertility preservation [182]. One questionnaire survey suggests that patients’ fertility preservation decisions are positively related to their wish to conceive (odds ratio (OR) 10.8, 95% confidence interval (CI) 3.5–34.4) and negatively associated with the expected burden of fertility preservation treatments (OR 0.08, 95% CI 0.02–0.3) [183]. In addition, fertility-related psychological distress is prevalent and persistent in cancer survivors and can reduce the quality of life [184].

## 6. Conclusions

Gonadotoxic chemotherapy such as alkylating agents can result in iatrogenic POI and loss of fertility in prepuberal girls and young women with cancer. To prevent loss of ovarian function and fertility in women with cancer, individualized strategies including established and experimental techniques should be provided based on patient age, marital status, economic status, chemotherapy regimen, cancer type, staging upon diagnosis and the possibility of treatment delay. Effective multidisciplinary oncofertility strategies that involve a highly skilled and experienced oncofertility team which consists of medical oncologists, gynecologists, reproductive biologists, oncologic surgeons, patient care coordinators, psychologists, and research scientists are necessary to provide cancer patients with high-quality care.

## Figures and Tables

**Figure 1 ijms-21-07792-f001:**
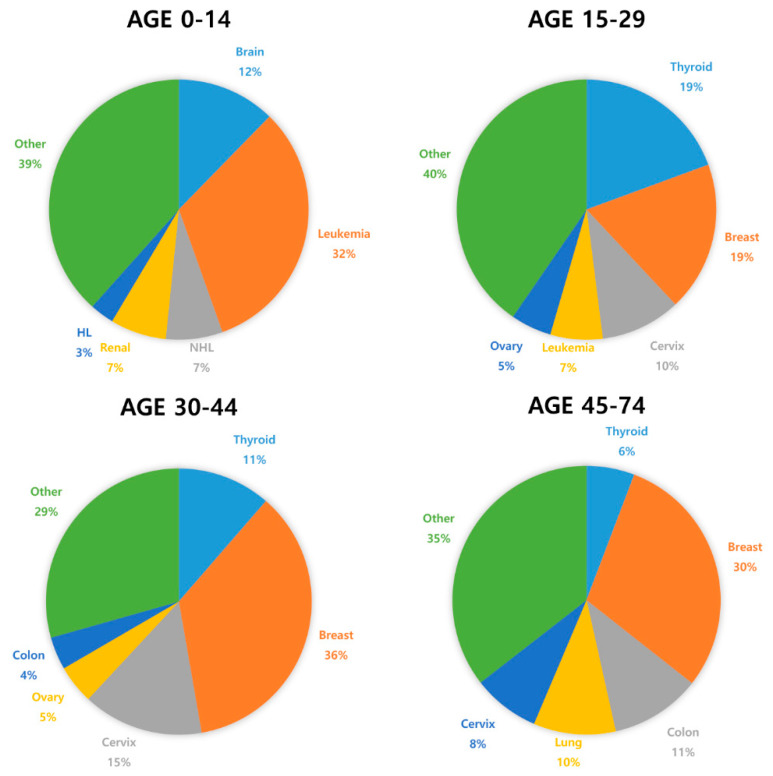
The most commonly diagnosed cancer types in females by age worldwide (2018) [27].

**Figure 2 ijms-21-07792-f002:**
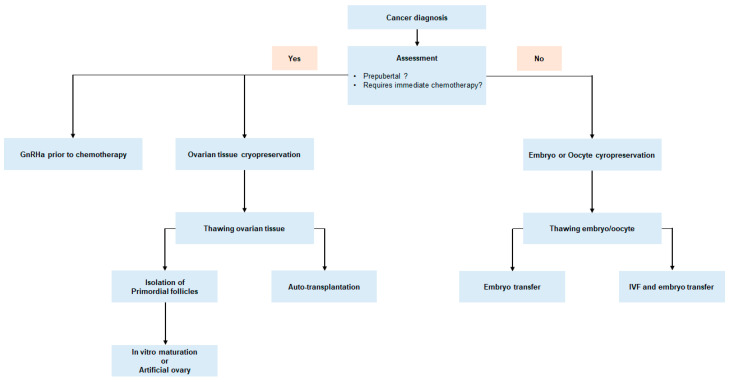
Fertility preservation approach for women with newly diagnosed malignancy.

**Table 1 ijms-21-07792-t001:** Common malignancies occurring in prepubertal girls and women at reproductive age and the risk of chemotherapy-induced gonadotoxicity.

Diagnosis	Chemotherapy Protocol	Risk of Iatrogenic POI
Non-Hodgkin lymphoma	Cyclophosphamide, hydroxydaunorubicin, oncovin, and prednisone (CHOP) (four to six cycles) Rituximab, cyclophosphamide, hydroxydaunorubicin, oncovin, and prednisone (R-CHOP) (four to six cycles)	<20% [3,28]
Hodgkin lymphoma	Adriamycin, bleomycin, vinblastine, and dacarbazine (ABVD)	<20% [3]
	Mustargen, oncovin, prednisone, and procarbazine (MOPP)	10–50% [29,30,31,32]
	Bleomycin, etoposide, adriamycin, cyclophosphamide, oncovin, procarbazine, and prednisone (BEACOPP) (eight cycles)	50–95% (age dependent) [33]
Acute lympho- cytic leukemia	Most standard chemotherapy protocols do not include a gonadotoxic multi-agent	<20% [3,13,14]
Acute myeloid leukemia	Most standard chemotherapy protocols do not include gonadotoxic anthracycline/cytarabine	<20% [3,13,14]
Breast cancer	Cyclophosphamide, methotrexate, fluorouracil (CMF) (six cycles) Cyclophosphamide, epirubicin, fluorouracil (CEF) (six cycles) Cyclophosphamide, eoxorubicin (adriamycin), fluorouracil (CAF) (six cycles)	>80% [3] (≥age 40)
		30–70% [3] (age 30–39)
	Doxorubicin (adriamycin), cyclophosphamide (AC) (four cycles)	30–70% [3] (≥age 40)
		>20 [3] (age 30–39)
Others	Cyclophosphamide ≥ 7 g/m^2^ in females < 20 years Cyclophosphamide ≥ 5 g/m^2^ in females > 40 years Any alkylating agent (e.g., cyclophosphamide, ifosfamide, busulfan, carmustine, lomustine)	>80% [34,35]
	Cyclophosphamide ≥ 5 g/m^2^ in females 30–40 years	30–70% [34,35]
	Taxanes Oxaliplatin Irinotecan Monoclonal antibodies (trastuzumab, bevacizumab, cetuximab) Tyrosine kinase inhibitors (erlotinib, imatinib)	Unknown

**Table 2 ijms-21-07792-t002:** Summary of major options for female fertility preservation and restoration after chemotherapy.

		Success Rate	Special Considerations
Established options	Embryo cryopreservation	Pregnancy rate of 30–40% per embryo	● Ovarian stimulation is not an option in prepubertal girls ● Ovarian stimulation takes several weeks ● Embryo freezing may be refused by unmarried women who do not want sperm donation
	Egg cryopreservation	Pregnancy rate of 4.5–12% per oocyte	● Ovarian stimulation is not an option in prepubertal girls ● Ovarian stimulation takes several weeks ● Suitable for unmarried women who do not want sperm donation
Experimental options	Ovarian tissue cryopreservation and auto-transplantation	Pregnancy rate of 20–40% per transplantation	● Can be performed in prepubertal girls or women who do not have enough time before chemotherapy ● Endocrine function may be restored after transplantation ● Spontaneous conception may be possible after transplantation ● Ovarian tissue transplantation should be contraindicated in women with primary or metastatic ovarian cancer ● Surgery is required to obtain tissue
	Oocyte in vitro maturation	Unknown	● Can be performed in prepubertal girls or women who do not have enough time before chemotherapy ● Safer than ovarian tissue cryopreservation and auto-transplantation
	Artificial ovary	Unknown	● Can be performed in prepubertal girls or women who do not have sufficient time before chemotherapy ● Suitable for patients with premature ovarian insufficiency
	Stem cell technologies	Unknown	● May become an option for prepubertal girls Surgery is required to obtain tissue
Unknown	GnRH analog	Debatable	● May be the only option when immediate cancer treatment is needed ● May protect ovarian function ● Unproven efficacy

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
