# Peer review of "Advances in the Treatment and Prevention of Chemotherapy-Induced Ovarian Toxicity"

_ijms, 2020, doi:10.3390/ijms21207792_

Round 1
Reviewer 1 Report
Dear authors,
Thank you for your nice job to revise the manuscript. The authors could revise the manuscript well. There are some minor points to revise.
Please do not use Korean in comment boxes.
Figure 1
AGE 30-45 should be AGE 30-44.
Reviewer 2 Report
Most of the points raised by the reviewer have been addressed satisfactorily. Authors have edited and improved the manuscript. Moreover they have added additional figures which nicely compose with the manuscript.
This manuscript is a resubmission of an earlier submission. The following is a list of the peer review reports and author responses from that submission.
Round 1
Reviewer 1 Report
Dear authors,
The authors made a great work to review recent advances in the treatment and prevention of cytotoxic chemotherapy-induced ovarian toxicity. This manuscript is well-written. However, there are a lot of carbon copy sentences. It is not recommended to make a carbon copy narrative review. Please revise the entire manuscript before your next submission.
Best,
Author Response
Author's Notes
Reviewer 2 Report
REVIEWER COMMENTS:
In this manuscript, Hyun-Woong and colleagues present a review regarding treatment and prevention of chemotherapy-induced ovarian toxicity. The review holds some interest, but has several analytical flaws that must be addressed. A point-by-point critique of the paper follows:
1. Both presented tables are unclear and difficult to follow. In Table 1. chemotherapy regimens are presented in a chaotic manner. Many abbreviations are used, without proper information anywhere near the table, making it difficult to understand. Furthermore, authors provide the following information „CHOP 4-6 cycles”; what if fewer or more chemotherapy cycles were used? Comments concerning not only the type of chemotherapy but also amount of cycles and protocol duration should be added. The table would be easier to follow if cancer types treated with according chemotherapy protocols were stated and listed accordingly to its risk of gonadotoxicity. Also, not all chemotherapy protocols with high ovarian chemotoxicity potential are listed, ex. BEACOPP for Hodgkins Lymphoma.
2. Table 2 should be remodeled. It would be better to begin with pregnancy success rate in the first column starting with the highest pregnancy success rate as this is one of the most important factors when choosing the possible treatment option. The methodology is unnecessary in the table as it is furtherly explained below in the following paragraphs.
3. There are several factors that influence decision making on choosing the oncofertility treatment option. Given the multi-drug treatment protocols, cancer types and staging upon diagnosis, many characteristics should be considered upon deciding on the options of fertility preservation. Also type of cancer (hematological vs solid tumors) should be evaluated as cancer cells may be present in potential oocyte tissue and therefore fertility preservation plans may be different. Moreover authors should further elaborate on the possible fertility preservation plans in hormone receptor-positive breast cancer and possible usage of GnRH analogues.
4. One of the most important problems is poor patient accessibility to oncofertility programs. Having a more advanced stage of cancer, early start of chemotherapy treatment is usually required, resulting in a low amount of time to consult with specialists and conduct fertility preservation. This should be raised in the article by proposing adequate fertility preservation options for specific patient groups. Financial costs should be considered as one of the potential problems with therapy accessibility problems.
5. Pros and cones regarding embryo vs oocyte conservation should be evaluated; there is no comment on the possible problems with ICSI and sperm injection failure, nor the effect of long term oocyte/ embryo storage on possible pregnancy rate and success once the patient is ready and willing to start family planning. As children, adolescents and young adults who underwent cancer treatment are the most probable group of patients that may be affected by infertility, long time of material preservation may be a possible issue.
6. Furthermore authors do not comment on potential problems with ovarian stimulation protocol that may result in a delayed start of chemotherapy due to possible adverse effects (ovarian hyperstimulation syndrome, complications during oocyte retrieval, anesthesia). As chemotherapy is often required to be started immediately, only one cycle of ovarian stimulation is often possible. This may result in a low count of mature high-quality oocytes retrieved from the patient.
7. Even if there is no early ovarian toxicity, many female patients experience early menopause. Authors did not comment on any possible timing of family planing nor patient follow-up post chemotherapy treatment.
8. As the decision about fertility preservation is often very emotional, authors should evaluate the psychological aspects that influence patients choice. Pregnancy success rate is certainly one of them, that is why it should be emphasized. It would be also potentially interesting to mention which fertility treatment options are most commonly chosen and why.
9. Finally, there are some grammar errors. Text should be rechecked.
Author Response
Author's Notes
Reviewer 3 Report
The manuscript provides a review of the state of the art of fertility preservation during chemotherapy for cancer patients. It provides a short and concise summary of the options women have for preserving their fertility before or during chemotherapy, what gaps in our knowledge there is, and the risks of current chemotherapy regimens. Focus is placed on the evaluation of individual patient needs and their situation, as well as the potential options patients have.
If word limits permit, a few further details (some listed below) would be helpful for readers and further enhance the manuscript. Authors could go into more detail about the details of the various methods and be a bit more comprehensive overall, although the reference list is very thorough.
It seems that the authors’ sections are a bit disordered. It looks like sections should be 3.3.1 and 3.3.4 should be sections 3.3 to sections 3.7, whereas section 3.3.5 should be section 4.
Is this statistic worldwide? “Over 150,000 reproductive-aged individuals face fertility-threatening cancer treatments each year”
It would be interesting if authors could give their opinion on how many hospitals or clinics have the available expertise to have “oncofertility teams”.
In the first paragraph of the Introduction, it would be useful to readers to also mention what other adverse side effects occur secondary to fertility loss (altered risk of reproductive/breast cancers, cardiovascular and metabolic risks, and sometimes muscular changes).
It would be very interesting if authors included some type of graph on age at which cancer is diagnosed in girls/women.
Authors should think about rephrasing sentences that are repeated in the abstract and other sections.
Is the burnout effect understood or proven in animals better than in people? A bit more detail would be useful. Knocking out or actively inducing the mentioned proteins is known to cause follicle activation in mice, certainly.
Line 270, it would be more useful if authors give a number or percentage of patients who do something about their fertility?
Table 2:
For ovarian transplantation, isn’t surgery always required? Authors say “May require surgery to obtain tissue”
The way the authors list the GnRH treatment seems biased: “debatable option”. I would suggest saying “unknown”. Ovarian stem cells are also highly debated.
It would be good to include the phrase ovarian suppression while talking about GnRH analog treatments. Authors state: “However, GnRH agonists should not be used if proven fertility preservation methods are available.” Are GnRN agonists really contraindicated or are the effects just not proven to work? Could multiple strategies be employed in sequence if the patient has enough time? I’m guessing they cannot be used in combination with ovarian stimulants, though that was not stated here.
